# Cooperative Parent-Mediated Therapy in Children with Fragile X Syndrome and Williams Beuren Syndrome: A Pilot RCT Study of a Transdiagnostic Intervention-Preliminary Data

**DOI:** 10.3390/brainsci12010008

**Published:** 2021-12-23

**Authors:** Paolo Alfieri, Francesco Scibelli, Laura Casula, Simone Piga, Eleonora Napoli, Giovanni Valeri, Stefano Vicari

**Affiliations:** 1Child and Adolescent Psychiatry Unit, Department of Neuroscience, Bambino Gesù Children′s Hospital, IRCCS, 00165 Rome, Italy; francescoscibelli@gmail.com (F.S.); laura.casula@opbg.net (L.C.); eleonora.napoli@opbg.net (E.N.); giovanni.valeri@opbg.net (G.V.); stefano.vicari@opbg.net (S.V.); 2Clinical Epidemiology, Bambino Gesù Children’s Hospital, IRCCS, 00165 Rome, Italy; simone.piga@opbg.net; 3Department of Life Sciences and Public Health, Catholic University, 00168 Rome, Italy

**Keywords:** parent-mediated therapy, RCT, autism spectrum disorders, fragile X syndrome, William Beuren syndrome, transdiagnostic intervention

## Abstract

Children with fragile X syndrome and William Beuren syndrome share several socio-communicative deficits. In both populations, around 30/35% of individuals meets criteria for autism spectrum disorder on gold standard instruments. Notwithstanding, few studies have explored feasibility and validity of therapy for socio-communicative deficits in individuals with these genetic conditions. In this study, we present preliminary data on a pilot RCT aimed to verify the effectiveness of cooperative parent-mediated therapy for socio-communicative deficits in a transdiagnostic perspective in a small sample of 12 participants. Our preliminary data showed that the experimental group had significant improvement in one socio-communicative skill (responsivity) and in clinical global impression, while the control group in an adaptive measure of socialization and word production. Implications of these results are then discussed.

## 1. Introduction

Autism spectrum disorder (ASD) is a heterogeneous complex neurodevelopmental disorder characterized by socio-communicative deficits and restricted, repetitive patterns of behaviors and interests [1]. ASD occurs in approximately 1:100 in the world [2] and in 1:54 in USA [3]. In approximately 10–15% of cases, ASD shows some genetic conditions [4,5], including tuberous sclerosis complex, fragile X syndrome (FXS), Down syndrome, Angelman syndrome, Rett syndrome, and, most recently considered, William Beuren Syndrome (WBS) [6,7,8,9,10].

FXS is the most known inherited form of intellectual disability, and it is considered the most common monogenic cause of ASD [11], accounting for an estimated 1% to 6% of all cases of ASD [12]. The estimated prevalence of FXS is ~1/4000 to 1/5000 in males and ~1/6000 to 1/8000 in females [13,14]. FXS is due to an expanded CGG repeat sequence (>200 repeats), termed “full mutation,” in the 5′ untranslated region of the FMR1 gene located at Xq27.3. Most males with full mutation have mild to moderate Intellectual Disability (ID), while the phenotype of females is considered milder (one third of females with the full mutation have normal intellectual function) [12]. A large portion of individuals with FXS shares difficulties in socio-communicative skills, such as modulation of non-verbal communication and difficulties in relationships with peers, as well as social withdrawal, and repetitive and restricted behaviors and interests [12,15]. Studies on males with FXS have showed that 30% to 54% met criteria for ASD, while this percentage is reduced in females (16% to 20%). Investigation conducted with gold standard instrument showed that around 35% of males met criteria for ASD on ADOS [16].

WBS is a rare genetic disorder (1:7500) [17] caused by a de novo hemizygous microdeletion on chromosome 7q11.23. The deletion usually ranges from 1.55 Mb (95% of cases) to 1.84 Mb (5% of cases) [18]. Individuals with WBS usually have global developmental delay/intellectual disability, with relative spared language skills and markedly impaired visuospatial skills [19,20,21]. Most individuals have mild to moderate ID, with an intelligence quotient (IQ) typically ranging from 40 to 90. The cognitive profile is uneven, with notable impairments in visuospatial processing skills but preserved expressive language and facial processing skills. Children with WBS show a distinct social-affective profile characterized by exaggerated sociability with strangers, an increased frequency of affective prosody, strengths in face perception and face recognition memory, and an increased interest in music, especially in the rhythm and emotional flavor of music [22]. The social profile of people with typical deletions (1.5 Mb) moreover is characterized by alterations in social judgment, emotional processing, theory of mind, and disinhibition. Furthermore, this social profile is related to neuroanatomical and neurophysiological alterations in the amygdala, fusiform gyrus, and orbitofrontal and parietal cortices. These cortical areas have been associated with the social cognition domains of emotional processing, social judgment, theory of mind, and empathy [23,24].

However, one of the most renowned features in individuals with WBS is the hyper-sociability and interest in people. For that reason, WBS has been considered for a long time a disorder that is the “polar opposite” of ASD [25]. Notwithstanding, recent studies have showed more similarities than differences between ASD and WBS. Children with WBS showed some deficits in socio-communicative abilities during preschool age (difficulties in pointing, gestures, giving objects, showing objects, eye contact, initiation and response to joint attention, and integration of eye contact with other non-verbal behaviors) as well as during school age (deficit in social relationships, pragmatic language, and emotional awareness) [8,21]. More than being opposite conditions, these authors suggests that WBS and ASD seems to have an opposite pattern of social motivation (enhanced in WBS and weakened ASD) and a common pattern of social cognition. Furthermore, results from cross-syndromes studies on adaptive level showed that individuals with ASD and WBS, when matched for age and cognitive level, were globally similar. Main differences emerged only in preschoolers in communication [26] and socialization [27] domains, while these differences were not found in school-aged children. Moreover, an investigation on communication adaptive profiles showed that expressive skills where higher than mental age, while receptive skills were found to be significantly lower, thus indicating some deficit in comprehension skills, including pragmatic abilities [20]. Finally, around 30–35% reached criteria for ASD [8,9,10] on gold standard tools for autism diagnosis (ADOS/ADOS-2). 

Thus, individuals with FXS and WBS show several socio-communicative deficits. In both populations, around 30/35% of individuals met criteria for ASD on gold standard instruments. For this reason, several authors have suggested testing feasibility and validity of therapy for ASD in children with FXS and WBS [12,21]. Notwithstanding, there is still a dearth of research in this field [28,29,30,31] especially for individuals with WBS. As far as we know, no studies have been conducted to explore feasibility and validity of therapy for socio-communicative deficits in individuals with WBS. Furthermore, most of the studies conducted on FXS have tested therapies or strategies by using single subject designs. The lack of research on intervention has brought parents to rely on therapies with few evidence of effectiveness [31].

Parent-Mediated Therapy (PMT) is a group of “technique-focused interventions where the parent is the agent of change, and the child is the direct beneficiary of treatment” [32]. PMT is commonly used in preschoolers with ASD. PMT has demonstrated increasing evidence of effectiveness in improving socio-communicational skills of children with ASD in several randomized controlled trials (RCT) [33,34,35,36,37,38]. Some recent researchers have extended the use of PMT to children with genetic disorders and autistic features, such as FXS [31]. While showing encouraging results, the research showed some limits, such as small sample size (four parent–child dyads), the use of single subject design, and the absence of control group. Other studies [39] have used “parent implemented intervention”. However, as far as we know, this kind of intervention was focused merely on language development, while PMT is usually focused on wider socio-communicative skills such as joint attention, emotional regulation, socio-emotional social engagement, social motivation, synchrony and sensitivity, imitation of gestures or objects, verbal and non-verbal communication, functional and symbolic play, social routines and anticipations, cooperative interaction, functional communication, conversation and cognitive flexibility, as well as promotion of language development. 

In this study, we present preliminary data on a pilot RCT aimed at verifying the effectiveness of cooperative PMT (CMPT) for socio-communicative deficits in children with FXS and WBS. CPMT is a parent coaching intervention focused on the ASD core symptoms. CPMT belongs to the group of Naturalistic Developmental Behavioral Interventions (NDBI) [40] with a specific focus on the promotion of cooperative interactions [41,42,43,44,45,46]. The aim of CPMT is to help parents promote seven target skills in their children: socio-emotional engagement, emotional regulation, imitation, communication, joint attention, play and cognitive flexibility, and cooperative interaction. In a previous RCT [38], CPMT has showed evidence of short time effectiveness in improving socio-communicational skills, ASD symptom severity, and in reducing emotional problems as well as parental stress related to parent–child dysfunctional interaction.

We chose to test the effectiveness of CPMT in children with FXS and WBS in a transdiagnostic perspective [47]. Currently, a relevant topic in the discussion of neurodevelopmental disorders is whether to consider a categorical (e.g., DSM 5 diagnoses) or dimensional perspective (e.g., studying different dimensions, such as socio-communicative skills) [47]. Furthermore, the relationship between the phenotype of genetic syndromes and ASD is highly controversial. Some large cohort studies [48] state that ASD is a unitary and categorical phenotype strongly influenced by the presence of a single specific causal factor, while others [49] propose that ASD is a multi-faceted construct where causal factors have variable and complex interactions. For example, some authors [50] have proposed a multidimensional model for understanding the structure of autism symptom phenotype, identifying three main dimensions: social-communication, inflexible language and behavior, and repetitive sensory and motor behaviors.

In this study, we have decided to use a transdiagnostic framework, moving from a diagnosis-centered to a child-centered perspective. We have considered that dimensions (such as socio-communicative deficits), rather than categorial characteristics that form a particular diagnostic group, are the most impactful features for the child’s evolution.

Our hypothesis is that CPMT, in addition to conventional rehabilitation therapies (mainly speech therapy and occupational therapies), could contribute to the enhancement of socio-communicative skills, as well as to the reduction in emotional and behavioral problems in a small sample of 12 children with FXS and WBS. We also expected an improvement in parental quality of life and stress. Finally, we expect to find that participants would receive the best rating on a clinical general impression scale. 

## 2. Materials and Methods

### 2.1. Participants

Families were referred to the study by Department of Child and Adolescent Psychiatry or Department of Genetic of Bambino Gesù Children Hospital after routine medical visits; alternatively, families contacted authors after reading the description of the study on the hospital website. Inclusion criteria were: 

(1) Age between 1 and 7 years; 

(2) Molecularly confirmed diagnosis of FXS or molecularly confirmed diagnosis of WBS; 

(3) Socio-communicative deficits clinically detected during the first assessment and highlighted by the Socio-Conversational Skills Rating Scale (SCSRS)—Italian version [51], a measure of preschoolers’ conversational assertiveness and responsiveness;

(4) Score ≥ or = 4 in clinical global impression—severity [52]. 

The only exclusion criterion was the involvement of parents in a parent training during first assessment. Fifteen children met inclusion criteria. One family retired from the project after the baseline assessment because of family problems of organization. Fourteen children and their parents were involved in the study (RCT code: NTC04610424). Two out of fourteen participants dropped out during the study. All of them were allocated in the treatment group. Motivation for drop out was different: one family decided to interrupt participation because of the distance between their home and the hospital; one family had to interrupt the treatment because of social restriction due to COVID-19. Finally, 12 children completed the study, 7 children with WBS and 5 children with FXS. Demographic and clinical characteristics of the groups (7 child in the control group, 5 in the experimental group) are described in Table 1. Diagnoses were distributed as follows: 4 children with WBS and 3 with FXS in control group, 3 with WBS and 2 with FXS in the experimental one.

### 2.2. Procedure

During the first assessment a child psychiatrist evaluated if the child met inclusion criteria and explained the informed consent to the parent. Once parents decided to participate to the study an identification number was assigned to each participant, and he/she was evaluated through an assessment battery (see Measures). All children received a blinded assessment by a multi-professional team of child psychiatrists and clinical psychologists. All outcome measures were assessed at baseline (Time 0) and at post-treatment (Time 1) after 9 months (in mean) for both groups. All participants completed the intervention in 6 months. Children were randomized using a computer algorithm based on pre-specified blocks based on their diagnosis (FRX vs. WBS). Assessors and supervising research staff were independent and unaware of allocation to control/treatment groups. Families and therapists could not be blinded to treatment allocation. The allocation was revealed to assessors and supervising research staff only after completing the post-treatment assessment. The control group received treatment as usual (TAU) provided by National Health System (usually speech language therapy and occupational therapy); the treatment group received TAU plus CMPT. 

The hospital’s ethics committee approved the study (protocol number 1324_OPBG_2017) and parents provided written informed consent. All procedures were in accordance with the Helsinki Declaration. 

### 2.3. Measures

#### 2.3.1. Child Assessment

We decided to use to some “out of age range” tools in the measurement of socio-communicative and language abilities by considering the “developmental age” reached by our children and the presence of deficits in socio-communicative skills.

##### Socio-Communicative Skills

Socio-communicative skills were assessed through the SCSRS—Italian version [51] and Early Social Communication Scales (ESCS) [53]. The SCSRS (in Italian, Abilità Socio-Conversazionali del Bambino—ASCB [51] is a measure of preschoolers’ conversational assertiveness and responsiveness. This tool is composed of 15 items for assertiveness and 10 items for responsiveness. Parents were asked to rate items according to the perceived frequency of occurrence. Responsiveness contains three types of questions: “respond to question”, “respond to requests”, “keep contingency”. Assertiveness contains three types of questions: “make questions”, “make requests”, “proposals”. This tool was designed for children between 12 and 36 months. We used it considering the socio-communicative abilities of our population, described in the abovementioned literature.

ESCS is a videotaped semi-structured toy play interaction that requires between 15 to 25 min to be administered. ESCS is administered at a table with the child sitting in front of the experimenter. Several toys, such as wind-up toys, a puppet, a book, posters, and social toys (e.g., sunglasses, a comb, and a hat) were presented to each child. The experimenter waited for initiation of behavioral requests (such as give a toy to the experimenter to activate it), social initiation (such as give the ball to start a game), or joint attention initiation (such as indicate to show an active wind-up toy). Furthermore, the experimenter tried to elicit a response to the same ability (behavioral response, social response, and joint attention response). The original version was designed to provide measures of nonverbal communication skills in children between 8 and 30 months. This tool could be also used with children with developmental delays and communicational deficits that fall in this age range. A trained examiner administered the ESCS in accordance with procedures outlined in the ESCS manual [53], with some adaptation due to the characteristics of our sample (mainly the age and the presence of problem behaviors). Given that some of our children had severe problem behaviors, in some administration we used extrinsic reinforcers alternated with ESCS activities (i.e., soap bubbles) to allow these children to interact with the experimenter. Furthermore, given that some children were independently able to action some wind-up toys, we had to use other way to block the activation of the toy (such as use of duct-tape). We considered the following scores from the tool: “total frequency initiate joint attention (IJA)”; “ratio higher IJA/total IJA”; “higher response joint attention”; “total frequency initiate behavioral response (IBR)”; “ratio higher IBR/total IBR”; total response behavioral request passes”; “total initiate social interaction”; “total response social interaction”.

##### Language Skills

Primo Vocabolario del Bambino (PVB) is the Italian adaptation of the MacArthur-Bates Communicative Development Inventory—MB-CDI [54,55]. PVB is a parent report questionnaire developed to assess receptive and expressive language of children between 8 to 36 months. The PVB has two forms: “PVB—gesture and words” for children between 8 and 24 months, and “PVB—words and sentences” for children between 18 and 36 months. Again, we used it considering the language delay in most of our participants (no one saturated the questionnaire, receiving the maximum score in word production at baseline). We used both versions depending on the child’s age and level of language. For our purpose, we considered only the evaluation of word production. We used raw data.

##### Adaptive Functioning

To assess adaptive functioning, we used the Vineland Adaptive Behavior System—Second Edition (VABS-II), Survey Interview Form [56], a tool used for individuals from birth to 90 years and 11 months. VABS-II is a caregiver interview and obtains four domain scores: communication, socialization, daily living skill, and motor skills. Each domain is composed of 2/3 subscales. For our purposes we used three domains (communication, socialization, and daily living) as well as two subscales of communication domain (receptive and expressive). The VABS-II adaptive behavior composite and relative domains yield age-based standard scores (M = 100, SD = 15).

##### Emotional and Behavioral Problems

Emotional and behavioral problems were evaluated through Child Behavior Checklist 1 ½–5 [57] and Child Behavior Checklist 6–18 [57], a widely used assessment system that comprises items ranging from 0 to 2. Item scoring is provided for several syndrome scales as well as three total scales (externalizing, internalizing, and a total problem). For our purposes, we used the three total scales as well as a syndrome scale that overlaps between the two forms (withdrawn scale). This scale is especially interesting for individuals with socio-communicative deficits [58]. 

##### Clinical Global Impression

We used both the CGI-severity (CGI-S) at baseline and the CGI-global improvement (CGI-I) scales at post-treatment. The CGI-S evaluates the clinician’s impression of the current state of the patient’s condition in the last seven days. The examiner should consider his total clinical experience with that given population, assigning one of the following scores: 1 = normal, 2 = borderline ill, 3 = mildly ill, 4 = moderately ill, 5 = markedly ill, 6 = severely ill, and 7 = the most extremely ill. The CGI-I assesses the patient’s change after the end of treatment/control. The examiner should use the following scores: 1 = very much improved, 2 = much improved, 3 = minimally improved, 4 = no change, 5 = minimally worse, 6 = much worse, 7 = very much worse. 

##### Cognitive/Developmental Assessment

Each cognitive/developmental profile was evaluated to assure that the two groups did not show differences at baseline. Based on age, language, developmental level, and time of administration, we used the Leiter International Performance Scale Third Edition (Leiter-3) [59] or The Griffiths Scales of Child Development-III [60].

Leiter-3 was the first choice in assessing the cognitive level of children. The Leiter-3 is a tool for assessing nonverbal cognitive, memory, and attention abilities, designed to be administered to individuals without language skills between 3 and 90 years. For our purpose, we used the non-verbal intelligent quotient (hereinafter IQ). When children did not complete the Leiter-3 because of problem behaviors or other noncompliance problems, we used the Griffiths Scale of Child Development-III. This scale is an assessment system for developmental level for children between 0 and 5 years and 11 months of age. The scale evaluates five domains of development: foundations of learning, language and communication, eye and hand coordination, personal–social–emotional, gross motor, and provides also a global score of developmental level (developmental quotient, DQ). 

#### 2.3.2. Parent Assessment

##### Parenting Stress

The Parenting Stress Index—Short Form [61] is a parent report questionnaire used to assess the stress level of both parents. PSI-SF comprises 36 items divided in three subscales: parent distress (PD), parent–child dysfunctional interaction (P-CDI), and difficult child (DC), plus a total PSI-SF score. For our purpose, we considered only the three subscales.

##### Quality of Life

The WHOQOL-BREF [62] was used to assess the quality of life of both parents. The questionnaire has 26 questions, scored into 4 domains: physical health (7 items), psychological (6 items), social relationships (3 items), and environment (8 items). The domain scores are transformed into scores between 0 and 100. Higher scores indicate better quality of life. Demographic information on educational level and employment of parents were obtained from this questionnaire.

### 2.4. Interventions

#### 2.4.1. Treatment as Usual

All 12 children received the TAU (speech language therapy and occupational therapy) provided as usual by National Health Services.

#### 2.4.2. Cooperative Parent-Mediated Therapy—CPMT

The CPMT is an NDBI parent coaching program that has already been adopted in the Children and Adolescent Mental Health Services of the Italian National Health System and in Child Neuropsychiatry Units of the Bambino Gesù Children’ Hospital. CMPT has demonstrated evidence of effectiveness in improving socio-communication skills [38]. The target skills of CMPT are socio-emotional engagement, emotional regulation, imitation, communication, joint attention, play and cognitive flexibility, and cooperative interaction. Each target skill has an individualized treatment plan based on a checklist assessment completed by the therapist at the beginning of the intervention. Parents and their child followed the therapy for 15 sessions (each session last 60 min) for a total amount of 6 months; twelve core sessions (one session per week) were delivered in the first 3 months, followed by 3 monthly booster sessions. Each session had a specific focus, that was implemented thought live active coaching in association with live modeling, and video feedback. At the end of each session, a memorandum on each specific topic and homework were given to the parents. Parents were asked to work daily for at least 1 h per day at home with the child. Two clinical psychologists specifically trained in intervention in ASD administered the CPMT. The clinical psychologists were trained in CPMT through direct supervision and video analysis by the child neuropsychiatrist who had implemented CPMT (GV). Some adaptations for children with higher verbal competencies or for children with difficulties in emotional and arousal regulation were required in this project. However, the intervention did not substantially differ from the previous [38].

### 2.5. Statistical Analysis

All statistical analyses were performed using STATA, Statistical Software: Release 13 (StataCorp LP, College Station, TX, USA). The statistical significance was set at *p* < 0.05. The Shapiro–Wilk test was used to assess the normality of the data. Categorical variables were summarized by absolute frequencies and percentages, and continuous variables by median and interquartile range (IQR). To determine statistical differences between groups, the chi-square or Fisher’s exact test was used for categorical variables, while the Wilcoxon or Mann–Whitney test was used for continuous variables. Furthermore, the patients were divided into two groups according to randomization criteria (7 in the control group and 5 in experimental group). To determine statistical differences for demographic and clinical variables between groups at baseline, we used the chi-square test or Fisher’s exact test and a nonparametric equality-of medians test to compare characteristics of children with or without experimental treatment.

## 3. Results

A total of 12 patients were analyzed, 7 were enrolled in the control group and 5 in the experimental group. 

At baseline the two groups had no statistically significant differences for all variables considered in the children (age, gender, IQ/DQ, CGI-S, and SCRS scores) and in the parents (educational level and employment). 

The treatment group statistically significantly increased the Responsivity score of SCRS: median at time t0 = 34; median at time t1 = 39, *p* = 0.042. The overall clinical impression (improvement) also shows a statistically significant difference between the median score, in the control group median = 4 and in the experimental group median = 2, *p* = 0.030. 

In the control group, the production of words (PVB) increased over time, at time t0 = 26 and at time t1 = 111, *p* = 0.031. Moreover, in the control group socialization (VABS II) increased over time, at time t0 median score = 64.5 and at time t1 median score = 66, *p* = 0.035. All results are described in Table 2.

## 4. Discussion

Here we present preliminary data of a pilot RCT designed to evaluate additional benefits of CMPT [38] on TAU in children with two genetic conditions often characterized by socio-communicative deficits: FXS and WBS. 

Our preliminary data showed that children in the experimental group were significantly ameliorated in one socio-communicative skill (responsivity) and in clinical global impression, while children in the control group in adaptive measure of socialization and in word production. 

Our preliminary data are somehow encouraging in considering parent involvement in the therapy of children with FXS and WBS. Despite the sample being too limited to draw a conclusion, we have found significant enhancement in responsivity and in clinical global impression. Responsivity is particularly important in the pragmatic abilities of those children, in particular of those with WBS. Children with WBS have difficulties in receptive abilities and in accurately responding to communicational bids of other children or adults. Enhancement in respond to questions/requests and the ability to keep contingency could help those children in establishing relationships more congruently with children and adults. Furthermore, as individuals with FXS and WBS shared the presence of anxiety symptoms (especially social anxiety in FXS) often linked to difficulties in comprehension of social requests, the enhancement in responsiveness ability could contribute to reducing these difficulties and thus social anxiety. However, this hypothesis is merely speculative and further research is requested to explore it.

There is a noticeable presence of a significant clinical global improvement in the experimental group when evaluated by clinician versus the absence of a significant improvement in other significant areas (such as scores in ESCS or emotional and behavioral problems). Even if we must consider that this could be an effect of limited sample size, we believe that other features not measured in this study could have influenced the change in the experimental group on clinical global impression. Personal communications with the therapists that implemented the CPMT as well as with the blinded experimenters revealed that children in the experimental group have more arousal and emotional regulation in social interactions at the end of the CMPT as well as more contingency to bids of communication offered. 

These clinical considerations could offer new insights into common and distinct characteristics of socio-communicative deficits in individuals with FXS and WBS. While individuals detected with “idiopathic” autism could have a primary deficit in socio-communicative skills, those with “syndromic” autism could have a secondary deficit in socio-communicative skills, mainly due to difficulties in regulation of arousal states and emotional activation that lead them to be unsynchronous in socio-communicative exchanges with other people. However, this hypothesis is merely speculative and must be confirmed by further studies exploring the role of arousal and emotional regulation in the expression of socio-communicative deficits. 

Concerning the significant differences in control group, we explained these results by considering the lowest level of socialization and word production at baseline in the control group when compared to treatment group. Control group had (in mean) only 26 words at baseline, and it is possible that TAU allows a late “explosion of vocabulary”. At post-treatment, the control group had quadrupled their vocabulary (111 words). However, the treatment group had five times more words (517) in terms of the absolute number of words. Thus, it is plausible that the significant effect in control group vs. experimental group came from differences at baseline. Similarly, adaptive socialization level in the experimental group was higher than the control group. So, it is plausible that a similar effect has occurred. However, we cannot exclude that TAU has a real significant effect on those variables. 

No differences were found in parental measures of quality of life and stress, thus indicating that CMPT does not seem to lead to significant changes in this area. Several studies have showed that other factors could contribute to improved quality of life and stress in families of individuals with developmental disabilities, such as the presence of an adequate support network [63] or specific therapies on parental cognition (e.g., mindfulness) [64]. These results suggest that if we want to improve quality of life and stress in these families, we must target an intervention separate from CPMT [65].

One interesting feature of this study is the choice of a dimensional perspective and transdiagnostic intervention framework. Our interest was to test whether a low-intensity (15 h over 6 months), and therefore potentially sustainable PMT could improve a dimension (socio-communicative skills) which frequently represents one of the most impactful areas in the psychosocial functioning of children with FXS and WBS, as well as in many other neurodevelopmental disorders. The use of a transdiagnostic framework allows a shift from a diagnosis-centered to a child-centered perspective. A dimension such as socio-communicative deficits, rather than categorial characteristics that form a particular diagnostic group, seems to have the greatest impact on psychosocial functioning of a child. Given the lack of studies on evidence-based therapies for children with FXS or WBS, the empirical verification, with methodologically rigorous studies, of the effectiveness of interventions on these clinically relevant dimensions seems currently necessary.

Summing up, preliminary data and qualitative analysis have revealed that CPMT could have a short-term enhancing effect in conversational responsiveness and in general clinical improvement. Furthermore, we cannot exclude that TAU alone could have a significant effect on word production and socialization. As far as we know, this is the first RCT exploring the effect of a therapy for ASD in a population of individuals with a genetic condition. The paucity of research highlights the need for further study in this field. Furthermore, the absence of RCTs makes it difficult to compare our data with other studies. 

The extension of our research to a wider sample could allow further exploration and understanding of the role of socio-communicative abilities in the enhancement of clinical global impression. Furthermore, this study has offered qualitative consideration on the role of arousal and emotional regulation in expression of socio-communicative deficits in FXS and WBS. Future study could explore this hypothesis by comparing the presence of atypical social communicative behaviors and difficulties in emotional regulation more in depth.

The main limitations of these preliminary data are the small sample size and the unbalanced number of participants in groups. For different reasons, we have had two dropouts in the experimental group. All of them could have been avoided with the use of a telehealth system [31]. Taking into account also the rareness of these disorders, future studies should carefully consider the use of a telehealth system to deliver this kind of treatment and avoid high rates of dropout.

## 5. Conclusions

There are only a few studies (often with low methodological quality) that have explored the feasibility and effectiveness of psychosocial interventions for social-emotional or behavioral problems in people with genetic conditions, such as FXS and WBS.

In this single-rater randomized controlled trial, preliminary data showed that children with FXS and WBS who received CPMT plus TAU presented with a significant improvement in some socio-communicative skills (conversational responsivity) and in clinical global impression.

We chose to adopt a dimensional and transdiagnostic perspective, similar to the dimensional systemic neuroscience framework developed by the NIHM Research Domains Framework (RDoC) [66] because it allows a shift from a diagnosis-centered perspective to a child-centered one, with the possibility to identify effective therapeutic interventions on dimensions more than categories, such as socio-communicative difficulties, that most impair psychosocial functioning. 

Subsequent studies with larger samples will allow us to confirm these preliminary data as well as to study more accurately specific mediators of the therapeutic response. 

## Figures and Tables

**Table 1 brainsci-12-00008-t001:** Demographic and clinical differences between groups at baseline.

Variable	Control Group Median (IQR) (Min–Max Range)	Experimental Group Median (IQR) (Min–Max Range)	*p*-Value
Sex (male:female)	3:4	3:2	1.000
Age	4.9 (3.1–5.6) [3.1–6.3]	5.4 (4.9–6.1) [3.2–6.5]	0.329
IQ/DQ	49 (49–71) [49–78]	56 (53–76) [50–78]	0.160
Clinical Global Impression-Severity	5 (4–6) [4–7]	5 (4–5) [4–6]	0.669
SCSRS Assertivity	37 (27–51) [23–65]	41 (34–56) [30–68]	0.723
SCSRS Responsivity	18 (15–41) [14–56]	34 (24–37) [20–45]	0.258
Level of education (mother)	4 (3–5) [2–5]	5 (4–5) [4–6]	0.202
Employment (mother)	28.6% (Employed) 71.4% (Unemployed)	80.0% (Employed) 20.0% (Unemployed)	0.242
Level of education (father)	4 (2–5) [2–5]	5 (4–5) [4–5]	0.144
Employment (father)	85.7% (Employed) 14.3% (Unemployed)	100.0% (Employed) - (Unemployed)	1.000

Legend: IQ: intellectual quotient; DQ: developmental quotient; IQR: interquartile range; min: minimum; max: maximum. IQ is expressed in standard scores. Clinical global impression—severity is a 7 point Likert scale. Level of education was the following: 1. no diploma; 2. elementary school; 3. middle School; 4. high school diploma; 5. Bachelor and master 6. post master specialization.

**Table 2 brainsci-12-00008-t002:** Pre–post control and experimental groups.

	Variable	Control Group Median (IQR) (Min–Max Range)			Experimental Group Median (IQR) (Min–Max Range)		
		T0	T1	*p*-Value	T0	T1	*p*-Value
**SCSRS**	Assertivity	37 (27–51) [23–65]	53 (37–56) [34–67]	0.091	41 (34–56) [30–68]	65 (49–66) [39–67]	0.080
	Responsivity	18 (15–41) [14–56]	37 (25–41) [17–49]	0.237	34 (24–37) [20–45]	39 (30–43) [22–48]	0.042 *
**ESCS**	Frequence Initiate Joint Attention (IJA)	21 (0–30) [0–66]	30 (13–36) [10–85]	0.063	12 (11–30) [2–62]	48 (12–52) [5–62]	0.225
	Ratio Higher/Total Level IJA	0.14 (0–0.46) [0–0.75]	0.33 (0.21–0.47) [0.00–0.92]	0.203	0.31 (0.08–0.48) [0.00–0.50]	0.27 (0.20–0.33) [0.16–0.38]	0.715
	Higher Level Response Joint Attention	100 (0–100) [0–100]	87.5 (87.5–100) [0–100]	0.856	87.5 (75–100) [12.5–100]	100 (100–100) [37.5–100]	0.572
	Frequence Initiate Behavioral Request (IBR)	20 (15–38) [5–38]	29 (29–34) [19–39]	0.128	30 (24–31) [2–33]	32 (31–34) [31–37]	0.104
	Ratio Higher/Total Level IBR	0.37 (0.16–0.4) [0.14–0.71]	0.31 (0.24–0.34) [0.05–0.52]	0.499	0.36 (0.29–0.42) [0–0.5]	0.41 (0.34–0.48) [0.32–0.56]	0.686
	Responding Behavioral Requests	75 (18–100) [0.7–100]	100 (60–100) [56–100]	0.212	100 (0–100) [0–100]	100 (84–100) [0–100]	0.317
	Initiate Social Interaction	2 (0–2) [0–4]	3 (2–3) [0–3]	0.094	2 (2–4) [0–4]	3 (2–4) [2–4]	0.160
	Total Responding social interaction	24 (10–45) [8–46]	38 (10–43) [6–49]	0.612	26 (20–39) [17–49]	37 (32–46) [25–48]	0.144
**VABS II**	Communication	51 (32–62) [32–67]	50 (28–54) [21–62]	0.248	54 (52–57) [20–62]	53 (43–65) [20–65]	0.586
	Daily Living	58 (49–66) [48–66]	56 (50–68) [42–73]	0.916	62 (57–74) [37–77]	58 (52–65) [35–65]	0.080
	Socialization	64.5 (58–70) [58–83]	66 (55–72) [48–75]	0.035 *	73 (73–81) [42–82]	72 (61–74) [38–88]	0.588
	Expressive	6.5 (3–9) [3–9]	8 (7–9) [2–9]	0.665	7 (6–7) [3–7]	8 (8–8) [2–10]	0.131
	Receptive	3.5 (2–7) [1–10]	5 (3–6) [1–9]	0.831	7 (5–9) [1–11]	9 (2–10) [1–10]	0.891
**CBCL**	Withdrawal	63 (52–79) [50–85]	63 (52–79) [50–85]	0.914	62 (52–63) [51–67]	51 (50–60) [50–68]	0.225
	Internalizing Problems	62 (43–73) [29–79]	56 (45–71) [37–80]	0.459	62 (49–63) [33–69]	58 (37–62) [29–62]	0.131
	Externalizing problems	63 (51–67) [44–73]	57 (51–67) [48–73]	0.398	53 (50–60) [40–65]	57 (51–67) [48–73]	0.586
	Total problems	59 (51–74) [39–79]	57 (50–74) [43–77]	0.866	63 (50–66) [35–68]	59 (40–67) [37–67]	0.686
**PVB**	Word Production	26 (1–258) [0–661]	111 (39–425) [0–676]	0.031 *	295 (229–243) [0–664]	517 (287–642) [0–644]	0.586
**CGI-I**	Clinical Global Impression—Improvement	-	4 (3–4) [1–4]		-	2 (1–2) [1, 2]	0.030 *^,^°
**PSI-SF Mother**	Parental Distress	65 (10–75) [5–90]	55 (25–85) [12–95]	0.351	80 (25–80) [5–100]	85 (55–100) [5–100]	0.339
	Parent-Child Dysfunctional Interaction	75 (45–95) [10–95]	65 (30–95) [25–100]	0.307	45 (40–65) [10–100]	75 (75–80) [35–100]	0.174
	Difficult Child	95 (80–100) [25–100]	90 (70–95) [25–100]	0.193	40 (25–90) [10–100]	80 (25–95) [5–95]	0.891
**PSI-SF Father**	Parental Distress	60 (5–75) [1–90]	20 (5–45) [1–95]	0.344	50 (10–55) [1–100]	65 (25–85) [5–100]	0.174
	Parent-Child Dysfunctional Interaction	75 (30–95) [25–95]	60 (35–95) [5–100]	0.732	60 (20–85) [1–95]	70 (40–80) [35–95]	0.174
	Difficult Child	90 (35–100) [30–100]	95 (30–95) [10–95]	0.146	50 (15–95) [1–100]	30 (30–95) [5–95]	0.586
**WhoQol Mother**	Physical health	29 (26–30) [20–33]	27 (24–29) [23–29]	0.444	24 (24–25) [20–27]	23 (22–28) [19–30]	1.000
	Psychological health	23 (19–26) [16–29]	20 (18–22) [18–23]	0.149	18 (18–19) [15–23]	18 (15–19) [14–22]	0.160
	Social relationships	13 (12–14) [11–15]	12 (11–12) [10–16]	0.227	11 (10–13) [6–14]	10 (8–10) [7–12]	0.172
	Environment	26 (21–33) [20–36]	26 (24–31) [21–31]	0.796	27 (25–28) [21–28]	24 (21–32) [20–32]	1.000
**WhoQol Father**	Physical health	31 (30–32) [22–33]	29 (29–32) [27–33]	0.798	25 (24–29) [21–34]	25 (23–29) [18–31]	0.172
	Psychological health	25 (21–27) [17–28]	20 (18–22) [18–23]	0.669	20 (19–21) [17–25]	21 (20–23) [15–24]	0.786
	Social relationships	11 (10–13) [8–14]	13 (12–14) [11–15]	0.073	11 (10–12) [8–14]	9 (9–10) [9–12]	0.098
	Environment	29 (27–32) [23–36]	31 (27–33) [26–35]	0.444	28 (25–28) [24–29]	25 (25–26) [20–29]	0.174

Legend: IQR: interquartile range; min: minimum; max: maximum. *p* level ≤ 0.05 significant. Frequency IJA, frequency IBR, initiate social interaction, total responding social interaction are absolute frequency. Higher level response joint attention and responding behavioral requests are percentage of occurrence. Ratio higher/total level IJA and ratio higher/total level are ratio between the higher level behaviors and the total number of behaviors. CBCL scores are T scores. ASCB, PVB, and WhoQol are raw scores. PSI-SF are expressed in percentile. VABS II scores are standard scores. Clinical global impression. Improvement is a 7-point Likert scale. * *p* < 0.05; ° This significance refers to difference between group at post treatment on clinical global impression—improvement.

## Data Availability

The data presented in this study are available on request from the corresponding author. The data are not publicly available due to privacy restriction.

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
