# Peer review of "Cooperative Parent-Mediated Therapy in Children with Fragile X Syndrome and Williams Beuren Syndrome: A Pilot RCT Study of a Transdiagnostic Intervention-Preliminary Data"

_brainsci, 2021, doi:10.3390/brainsci12010008_

Round 1

Reviewer 1 Report

The article entitled “Cooperative Parent Mediated Therapy in Children with Fragile 2 X Syndrome and Williams Beuren Syndrome: a pilot RCT 3 study of a transdiagnostic intervention. Preliminary data” is a pilot study in which the authors presented preliminary data on the effect of Parent Mediated Therapy in FXS and WBS individuals. This study represents an interesting attempt to explore the feasibility and validity of therapy for socio-communicative skills in individuals with genetic conditions. The main findings of this study are the following: i) the experimental group improved in socio-communicative skill (Responsivity) and in clinical global impression, ii) control group in adaptive measure of socialization and in word production.

Having said that, I have substantial concerns about the methodological and statistical decisions made by the authors, and the potential impact that these have had on the interpretation and robustness of their findings. The main issue here concerns the conclusion about the significant difference found at t1 in clinical global impression between the two group (experimental and control). The authors reported and discussed it as a significant enhancement (line 358 and 447 etc..) in the Clinical Global impression - Improvement scale. This interpretation is completely misleading because there is no data about t0, i.e., before the treatment. So that, even if in the post-treatment the two groups were statistically different, it is impossible to clearly attribute this change to the effect of treatment because we did not know if the two groups were comparable in term of Clinical Global impression before starting the intervention. One may also hypothesize that the two groups were already different before the treatment. If the authors cannot exclude the lack of difference at t0, they cannot conclude the difference found at t1 was specifically due to the type of intervention adopted, that is the manipulation.

Author Response

We thank the reviewer  1 and 2 for their interest and comments about our study.

Reviewer 1

In accordance with his/her comments and suggestions, we would like to give this answer:

To assess the clinical global impression we have used both the CGI-Severity (CGI-S) at baseline and the CGI-Global Improvement (CGI-I) scales at post-treatment. The CGI-S evaluates the clinician's impression of the current state of the patient's condition in the last seven days. While the CGI-I assesses the patient's change after the end of treatment/control.

CGI-S was assessed at time t0: both groups show a comparable level with median score = 5 (Table 1) and there are no statistically significant differences (p-value = 0.669), therefore the groups are comparable as they are homogeneous at baseline.

CGI-I is an evaluation carried out at time t1, and shows an improvement in the group that followed the experimental treatment (Table 2), the median score of the scale decreased from 4 in the control group to 2 in the group experimental, this difference was statistically significant (p = 0.030).

Reviewer 2 Report

The manuscript describes a preliminary study on two disorders, and nicely present the findings, and properly discuss the results.

The findings are of interest, and the manuscript is overall very nice. 

I only have minor comments:

The background can be improved, adding references on Williams syndrome. References and reviews on the social behavior properties in Williams syndrome are lacking, as well as studies on cognitive impairment in the disorder. The authors should also expand in the background (and perhaps also in the discussion) the information on potential neuroetiology that may explain Williams syndrome deficits. For instance, whether neuroanatomical features and deficits in the disorder, taking place in grey and white matter brain regions responsible for social behavior, etc. 

The number of participants is low, and hence the authors need to better emphasize the limitations of this research and the interpretation of the results coming from only 12 participants. 
While they nicely emphasize this in the discussion, I suggest they add a comment on this issue also in the abstract/introduction. 

Author Response

We thank the reviewer  1 and 2 for their interest and comments about our study.

Reviewer 2

 In accordance with his/her comments and suggestions,  we have modified the following aspects:

  • References and reviews on cognitive profile, social behavior characteristics and   neuroanatomical features associated with  social profile in Williams syndrome, were added in the Introduction (Lines 54-66)

  • Regarding the limitation of our sample, we agree with the reviewer 2 and we have emphasized this issue in the abstract (Line 21) e in the Introduction (Line 142)